# Partial Discharge-Based Cable Vulnerability Ranking with Fuzzy and FAHP Models: Application in a Danish Distribution Network

**DOI:** 10.3390/s25113454

**Published:** 2025-05-30

**Authors:** Mohammad Reza Shadi, Hamid Mirshekali, Hamid Reza Shaker

**Affiliations:** SDU Center for Energy Informatics, Maersk Mc-Kinney Moeller Institute, The Faculty of Engineering, University of Southern Denmark, 5230 Odense, Denmark

**Keywords:** asset management, cable replacement, condition-based maintenance, fuzzy analytic hierarchy process, Monte Carlo simulation, partial discharge, Spearman correlation

## Abstract

Aging underground cables pose a threatening issue in distribution systems. Replacing all cables at once is economically unfeasible, making it crucial to prioritize replacements. Traditionally, age-based strategies have been used, but they are likely to fail to depict the real condition of cables. Insulation faults are influenced by electrical, mechanical, thermal, and chemical stresses, and partial discharges (PDs) often serve as early indicators and accelerators of insulation aging. The trends in PD activity provide valuable information about insulation condition, although they do not directly reveal the cable’s real age. Due to the absence of an established ranking methodology for such condition-based data, this paper proposes a fuzzy logic and fuzzy analytic hierarchy process (FAHP)-based cable vulnerability ranking framework that effectively manages uncertainty and expert-based conditions. The proposed framework requires only basic and readily accessible data inputs, specifically cable age, which utilities commonly maintain, and PD measurements, such as peak values and event counts, which can be acquired through cost-effective, noninvasive sensing methods. To systematically evaluate the method’s performance and robustness, particularly given the inherent uncertainties in cable age and PD characteristics, this study employs Monte Carlo simulations coupled with a Spearman correlation analysis. The effectiveness of the developed framework is demonstrated using real operational cable data from a Danish distribution network, meteorological information from the Danish Meteorological Institute (DMI), and synthetically generated PD data. The results confirm that the FAHP-based ranking approach delivers robust and consistent outcomes under uncertainty, thereby supporting utilities in making more informed and economical maintenance decisions.

## 1. Introduction

The worldwide aging of buried cable infrastructure poses a critical power system reliability issue, one that raises the risk of insulation failure and power outages. With approximately 10,000–12,000 km of lead-armored (APB) 10/20 kV cables installed primarily between the 1960s and 1970s, Denmark exemplifies the global issue of cable assets exceeding their designed lifespan of 30–35 years, beyond which their vulnerability to faults increases sharply [1]. Similarly, Sweden operates more than 437,000 km of underground cables, most of which have used cross-linked polyethylene (XLPE) insulation since the 1970s [2]. However, these cables can experience practical issues such as water tree formation, moisture ingress into the insulation, and other defects, all of which contribute to accelerated aging.

Several physics-of-failure models, such as thermal–oxidative aging, water tree growth, electrical tree propagation, and multi-stress Eyring models, have been developed to predict cable insulation degradation from laboratory-based offline testing. For example, the authors in [3] conducted thermal and corrosion degradation testing with the application of accelerated stress under controlled laboratory conditions on samples that underwent a destructive analysis. On the other hand, PD measurements, as standardized in IEC 60270, yield continuous, non-intrusive, and online condition monitoring, which renders PD particularly suitable and practical for utility-scale, network-wide cable management.

Although failures can occur due to various factors, PD remains one of the most recognized indicators of insulation degradation in aging cables. Today, utilities have initiated massive replacement schemes to upgrade old APB and XLPE cables, particularly those evidencing PD activity. However, due to the substantial capital investments and operational complexities associated with such replacements, cable section ranking based on the history of fault occurrences and PD characteristics is essential.

Conventionally, cable replacement has been a simple, age-based strategy, where the oldest cables are replaced first. But this is costly in large electrical networks with thousands of cable sections. Moreover, this replacement approach does not guarantee safety or reliability. A younger cable showing early signs of deterioration can pose a more immediate threat to network integrity than an older cable maintained in proper condition.

Much like human health, a cable’s longevity depends on its environment and operational factors such as soil composition, temperature, moisture levels, loading conditions, manufacturing quality, and installation procedures, which can either accelerate aging or preserve functionality beyond expectations [4]. This is why utilities are moving away from reactive guesswork and adopting proactive, condition-based approaches.

The prioritization of cable replacement, as well as awareness of the age of the cable, relies further on a perception of the historic fault statistics and activity of PD. The detection and analysis of PD activities, however, are technically challenging and depend on a good-quality sensor and successful interpretation of the signal. To support this, various PD measurement techniques have been developed for both offline and online applications. Electrical methods, such as the pulse current method [5], capacitive coupling [6], ultra-high-frequency (UHF) detection [7,8], and electromagnetic coupling [9], offer different trade-offs in terms of sensitivity, frequency response, and ease of installation. In addition, non-electric techniques like acoustic emission [10,11] and fiber optic [12] sensing offer utility where electromagnetic interference may hinder detection, and they are practical solutions to discharge source location. These techniques enable a more informed and targeted approach to cable condition monitoring and replacement scheduling [13,14].

Each PD measurement method has its own pros and cons. For example, ultrasonic sensing often struggles with signal loss and noise over long distances. To improve detection, newer systems place sensors along the cable to catch signals closer to where the problem occurs. These sensors are synchronized using LFM signals, which removes the need for GPS or fiber optics and still allows for accurate fault location [15]. Table 1 presents various sensors used for online PD detection in cables, along with their main advantages and limitations.

Meanwhile, machine learning is changing the way we understand PD signals. Instead of manually interpreting complex waveforms, models like 1D convolutional neural networks and graph-based algorithms can now learn the patterns hidden in noise, making it easier to identify the type of fault [16,17]. Alongside this, topological data analysis offers a new way of looking at PD signals. Visible patterns act to give a better indication of the slight differences between faults that would otherwise be missed, thus improving detection accuracy and reliability. However, as these AI models are often black boxes, explainable AI is essential to ensure transparency and trust [18].

**Table 1 sensors-25-03454-t001:** Overview of PD detection sensors [13,19].

Category	Sensor Type	Application Area	Frequency Range	Advantages	Disadvantages
Pulse Current Method	Coupling Capacitor	Internal discharges	50 Hz–1 MHz	Proven High sensitivity Widely used	Expensive Calibration needed Interference-prone
Capacitive Coupling	Pulse Capacitive Coupler	Surface discharges	100 kHz–1 GHz	Effective for surface PDs On-site usability No insulation contact	Limited detection range Calibration required Interference-prone
Electromagnetic Coupling	HFCT	All PD types	30 kHz–300 MHz	Wide frequency range Noise-resistant Shape-based detection	Cable-type dependent Requires circuit access High cost
Electromagnetic Coupling	Rogowski Coil	Transformer PDs	20 kHz–200 MHz	Easy installation Very high bandwidth	Lower sensitivity than HFCT Vulnerable to EM noise
UHF Detection	UHF Sensor	Cables, GIS, transformers	300 MHz–3 GHz	High sensitivity Enables PD localization	Close proximity required Complex setup
Radiometric Detection	Radiometric Sensor	Overhead lines, substations	30 MHz–3 GHz	Long-range detection Effective in open environments	External interference Requires line of sight
Electromagnetic Coupling	VHF Sensor	Internal cable discharges	30 MHz–300 MHz	Sensitive to cable PDs Clamp-on installation	Limited detection distance Affected by shielding
Acoustic Emission	Acoustic Sensor	Corona discharges	20 kHz–300 kHz	Immune to EM noise Quick and low cost	Weak signal detection Limited PD localization
Optical Sensing	Optical Sensor (UV/IR)	Surface and thermal PDs	UV/IR range	Real-time detection Immune to EM interference	Line of sight required Sensitive to ambient light
Distributed Temp. Sensing	DTS	All cable types	–	High spatial resolution Long-range coverage	High cost Affected by fiber stress

However, detection alone is not enough. Knowing the number of PD events, their severity, and their location gives us a clear picture of the cable’s health. This information enables us to prioritize cable sections based on risk. Furthermore, detailed information about the PD in each section allows for predictive maintenance planning, which is possible by predicting the oncoming failure and preparing for early intervention [20]. This further supports in-depth root cause analysis, which detects underlying causes, such as manufacturing defects or installation errors, and it improves design and material selection for future installations [21]. Also, leveraging this information enhances quality control and regulatory compliance programs that ensure that cables meet safety standards and performance standards.

Estimating the remaining life of cables using PD sensors has become an intriguing approach. In [22], researchers compared the measured PD charge with a set threshold to estimate cable life. For example, a 36-year-old cable showing a PD charge of 1394 pC against a threshold of 2500 pC was estimated to have about 28.6 years remaining using a linear model $\mathrm{RLT}=36\times \left(\frac{2500}{1394}-1\right)$. pC stands for picocoulombs, a unit that measures the tiny amount of electric charge released during a partial discharge event; specifically, one picocoulomb equals $10^{-12}$ coulombs. This charge is typically calculated as Q=I×t, where *I* is the discharge current, and *t* is the discharge duration. A higher discharge value in pC generally indicates more severe insulation stress or degradation. However, the behavior of PDs is non-linear, as insulation voids grow, and discharge amplitudes may increase exponentially over time. This behavior limits the reliability of life estimation models based on linear assumptions. Changes in weather and load conditions (winter and summer) might affect both the frequency and amplitude of PDs. In [23], the authors demonstrate that electrical stress influences PD number and amplitude. Higher stress leads to more frequent and intense discharges, which, in turn, promote faster insulation degradation. Here, the inclusion of the number of PDs and their severity provides a more accurate assessment of the cable’s vulnerability.

However, achieving such detailed PD monitoring is not without its challenges. Detecting PDs at nanosecond resolution naturally leads to large volumes of data. This, in turn, imposes enormous pressure on both the storage capacity and computational power. Current systems often capture the whole PD waveform and employ techniques such as wavelet transforms [5] or data-driven methods [24,25] to analyze this high-resolution signal. Although this comprehensive approach is meticulous, it raises questions about cost, storage, and processing. The main objective, however, is to provide early warnings of insulation degradation and assist in proactive asset management. Moreover, installing the same sensors in numerous critical underground cables in the network only aggravates these issues.

To overcome these issues, we are developing an innovative HFCT sensor that operates on analog data and uses targeted filtering to capture only the peak PD events over time rather than the entire signal. This sensor will be installed along the shield path at both ends of the cables. In this paper, we set aside sensor development details to concentrate on developing a comprehensive asset management framework. By generating synthetic PD data in PowerFactory, we simulate realistic cable degradation scenarios and assess cable risk based on the number and amplitude of PD events. Unlike conventional chronological age-based ranking methods that consider solely the age of the assets, our system presents a sophisticated solution that best describes the actual status and degree of the deterioration of every cable. Once our sensor is fully developed and deployed, real-world data will be integrated to validate and refine the framework, ensuring that it becomes a reliable and practical tool for predictive asset management in modern power networks.

Based on [26], cable replacement strategies are typically divided into seven categories, as presented in Figure 1. Ref. [27] proposes a model integrating preventive cable replacement into distribution network reinforcement planning by extending an automated low-voltage network planning tool. It uses a genetic algorithm to jointly optimize reinforcement and renewal actions to reduce redundant construction efforts. The authors in [28] present a data-driven model using Monte Carlo simulations to optimize underground cable maintenance and replacement cycles. The model identifies an optimal cycle of approximately 9.7 years that minimizes overall expenses by analyzing real failure rates and cost data. This approach offers greater numerical accuracy and realistic failure modeling. The study in [29] discusses how and when to replace or rejuvenate underground cables in a distribution network. By integrating different optimization techniques, including mixed-integer linear programming (MILP), dynamic programming, and TOPSIS, it seeks to achieve a balance between keeping costs low and improving reliability. Through a case study on a test network, the approach shows flexibility and practical value, even though it requires computational resources.

This paper adopts the condition-based replacement approach to prioritize cables according to their actual condition rather than relying solely on age or failure history. Our work is the first to use PD measurements as indicators of cable vulnerability and ranking. Since faults are still rare in the studied network, we propose a ranking framework that incorporates cable age, the number and amplitude of detected PDs, and the total number of end-users affected in the event of a cable failure. Due to the absence of an established ranking methodology for such condition-based data, we introduce two decision-making models, fuzzy logic and the FAHP, and compare their rankings with those of a traditional age-based approach. The proposed framework enables data-driven, expert-supported asset prioritization in the absence of failure history. It provides a structured and flexible means to translate condition indicators into actionable maintenance decisions.

In contrast, prior research has employed the FAHP in different contexts: the authors in [30] addressed expert uncertainty to rank node vulnerability; Ref. [31] used subjective linguistic assessments to evaluate renewable energy supply systems; and in [32], they applied the FAHP to systematically compare technical and economic criteria for assessing power system flexibility.

The data, collected from a Danish Distribution System Operator (DSO), include a ±10-year uncertainty in cable ages. To account for this, we apply a Monte Carlo simulation to generate more robust rankings. Given the nature of the data and the DSO’s practical replacement planning needs, fuzzy methods offer intuitive, expert-driven decision support despite their limitations in cost accuracy. Lastly, we perform a sensitivity analysis to evaluate the robustness of the proposed models under varying input assumptions.

The key contributions of this study include the following:Proposing a fuzzy logic and FAHP-based framework for cable vulnerability ranking.Utilizing real cable data from a Danish distribution network and meteorological inputs from DMI.Enabling a simplified and low-cost sensor design by using only peak PD values and event counts.Assessing robustness under uncertainty using Monte Carlo simulation and Spearman correlation.

The organization of this paper is as follows: Section 2 explains the envisioned methodology, including data preparation, the fuzzy logic system, and the FAHP framework. Section 3 presents the simulation scenario and results, including a ranking comparison and robustness analysis. Section 4 offers a discussion of the results, and Section 5 concludes the research with the findings.

## 2. The Proposed Methodology

This section delineates our methodological approach in two parts. First, we describe the network dataset and the procedure for generating synthetic PD data. Second, we detail the decision-making framework that integrates fuzzy logic and the FAHP to rank cables based on available network and PD data.

### 2.1. Dataset

This study utilizes a three-year hourly dataset from a real Danish distribution grid connected to a 10 kV station with 75 transformers, 75 lines, and 43 DGs (see Figure 2). The dataset comprises load profiles, DG outputs, network topology, line models, cable ages (with ±10-year uncertainty), and customer counts per cable, as provided by the Danish DSO. This ±10-year range accounts for uncertainty in installation records provided by the DSO, particularly for older cables installed before digital tracking systems were in place. While newer cables have more reliable service date information, many older ones lack precise documentation, making the uncertainty necessary for realistic modeling. The network is modeled in DIgSILENT PowerFactory. Since the sensor is still under development and real PD data are not yet available, we focus on establishing the ranking framework. Future research will involve experimental validation using field measurements once these sensors become operational. To create a synthetic PD dataset, we utilize three years of real network data, along with key environmental parameters, such as moisture level and soil temperature from the DMI [33], cable loading from hourly power flow simulations, and cable age. Seasonal variations are incorporated, as the number and amplitude of PD events differ between summer and winter.

Leveraging this multifaceted dataset that includes continuous environmental and electrical parameters, we apply a fuzzy logic framework based on the rule set in Table 2 to synthesize PD event data. This method enables us to emulate PD occurrences on each cable for 3 years. For explanation, high moisture infiltrates insulation and fosters the formation of water trees, while elevated soil temperatures accelerate chemical degradation and reduce dielectric strength, making insulation more susceptible to partial discharge events. After generating the PD probability values, we scale the normalized outputs to a range between 0 and 5 to represent the estimated number of PDs per day. Figure 3 illustrates the daily PD counts over three years for two randomly selected cables.

To simulate PD events, the model illustrated in Figure 4 is implemented in DIgSILENT PowerFactory and placed on the shields of underground cables. This setup allows us to replicate PD signals in the software environment, as exemplified in Figure 5 for measured PDs at both ends of a cable. Based on the estimated daily PD counts for each cable obtained in the previous step, synthetic PD signals are simulated, and the peak values of all simulated PD events are recorded for subsequent analysis. This approach reflects a practical measurement scenario, relying only on easily obtainable features such as event count and peak magnitude. By avoiding the need for complex waveform capture or high-frequency sampling, the method supports the development of simplified and cost-effective PD monitoring sensors. In total, 25,000 PD samples are simulated across the network over three years, proportionally distributed according to each cable’s estimated daily PD activity. Among the simulated lines, 33 cables exhibit detectable PD activity above the assumed sensor threshold of 100 pC, while the remaining lines show no measurable PDs within this detection limit. Based on [34], the severity of PDs can be categorized into three levels depending on the type of cable, as shown in Table 3: Within Safe Limits, Moderate Risk, and Critical Risk. These categories correspond to the range of the discharge magnitudes and guide condition-based responses. Since the synthetic dataset includes numerous PD events with varying amplitudes, we define five boundaries to represent increasing risk severity levels.

For each cable, we calculate a cumulative PD risk index that aggregates the amplitude severity of simulated PD events. Each PD is assigned a weight based on its boundary level (e.g., 0.2 × low risk, 0.3 × next level, and up to 1 × critical risk). This index provides a continuous and interpretable measure of the insulation condition.

### 2.2. Fuzzy Logic Framework

Fuzzy logic is a computational technique that is suitable for combining quantitative information with qualitative expert judgments. This makes it useful in uncertainty modeling in complex and real-world decision-making issues. Unlike classical binary logic, which focuses on discrete true/false outcomes, fuzzy logic lets variables have degrees of truth ranging from 0 to 1 in which nuances in human judgment and imprecise data are incorporated [36]. In our study, fuzzy logic is applied to prioritize underground cables for replacement using uncertain inputs such as estimated cable age, the frequency and amplitude of detected PDs, and the potential impact on end-users. These inputs (both quantitative and qualitative) do not conform to rigid thresholds or exact mathematical models. Instead, fuzzy logic employs intuitive linguistic rules (e.g., IF cable is old AND PD activity is high THEN risk is high) that result in smooth decision boundaries and enhanced interpretability. Thus, fuzzy logic makes the whole task of cable replacement planning more reliable and human-friendly. A fuzzy system includes three main steps:

#### 2.2.1. Fuzzification

In our study, raw values like the age of a cable, the number of PDs, and the cumulative PD risk index are all precise numbers. But in real-world decision-making, these values are not simply “good” or “bad” together. Fuzzification turns these exact values into degrees of belonging (membership) to intuitive categories like “low”, “medium”, or “high”. For instance, a cable that is 25 years old might not be fully “old”, but it is not exactly “young” either. Although the time of service is a valuable indicator, accelerated aging due to elevated operating temperatures and environmental stressors must also be considered for a more comprehensive evaluation. This partial belonging is captured through membership functions, which allow us to represent this uncertainty and overlap using smooth curves. Various types of membership functions are available in fuzzy systems, such as triangular, trapezoidal, and Gaussian functions, each offering different shapes and flexibility [37]. This study uses triangular membership functions to represent the fuzzy sets corresponding to the input variables, as they are simple, interpretable, and computationally efficient. A triangular membership function is mathematically defined as Equation (Equation 1):(1)$$\mu \left(x\right)=\left\{\begin{array}{cc} 0, & x\leq a\;\mathrm{or}\;x\geq c\\ \frac{x-a}{b-a}, & a<x\leq b\\ \frac{c-x}{c-b}, & b<x<c \end{array}\right.$$
where *a*, *b*, and *c* represent the lower bound, the point of maximum membership, and the upper bound of the fuzzy set, respectively. The membership function μ(x)∈[0,1] indicates the degree to which a specific input value belongs to the associated linguistic category. As shown in Figure 6, these five plots visualize how input values like the age of a cable and the number of PDs are turned into fuzzy terms such as “low”, “medium”, or “high”. Instead of treating these values as simply good or bad, this approach captures the shades of meaning in between, making them easier to work with in the rule-based reasoning process.

#### 2.2.2. Inference—Reasoning Like an Expert

Once the inputs are fuzzified, the inference engine evaluates a series of expert-defined if–then rules. For example, *IF a cable is “old” AND PD activity is “high”, THEN vulnerability is “high”*. This step mirrors how engineers assess risk by weighing multiple factors rather than applying rigid thresholds. The fuzzy system combines these conditions smoothly, considering parameters such as cable age, PD severity, and the number of affected customers to generate a fuzzy risk estimate. In this study, 37 fuzzy rules are constructed based on expert knowledge of cable degradation processes, PD phenomena, and their operating consequences.

#### 2.2.3. Defuzzification—Translating Fuzzy Logic into Actionable Scores

Once the fuzzy system has reasoned through all the rules, it gives a range of possibilities in the form of a fuzzy output. To make this information usable in real-world decisions, it needs to be turned into a number. This is where defuzzification comes in. This study uses the centroid or “center of gravity” method. It works by finding the balance point of the fuzzy output shape, giving us a single, representative value: the vulnerability score. This score helps utility providers rank the cables to determine which ones need maintenance or replacement first.

### 2.3. Fuzzy Analytic Hierarchy Process

The analytic hierarchy process (AHP) has a proven track record in determining the relative importance of various decision criteria by systematically breaking them down into pairwise comparisons. However, its reliance on strict numerical judgments overlooks the inherent vagueness of expert opinions. In parallel, fuzzy logic provides a flexible way to describe uncertainty using linguistic terms (e.g., moderately more important), but it can over-emphasize certain top-level criteria without a structured hierarchy, often called “parent node score distortion”. By merging the clarity of the AHP’s hierarchical comparisons with the flexibility of fuzzy logic, the FAHP offers a balanced solution. Actually, through the FAHP, decision-makers can address questions like “Which cables are most susceptible to failure under uncertain aging conditions?” with a method that is both rigorous and receptive to expert nuance.

#### 2.3.1. Representing Expert Opinions as Triangular Fuzzy Numbers

In the classic AHP, comparisons are strict (e.g., “A is 3 times more important than B”), but experts often use softer statements like “A is slightly more important”. Fuzzy logic captures this uncertainty by assigning each comparison a range. The FAHP allows the judgments to be expressed as triangular fuzzy numbers (TFNs), denoted by (l, m, u), where l is the lowest likely value, m is the most typical value, and u is the highest possible value. The following provides a better clarification:A judgment of “equally important” might be translated into (1,1,1).“Moderately more important” might be (2,3,4).If criterion A is only slightly less important than criterion B, the reciprocal fuzzy number $\left(\frac{1}{4},\frac{1}{3},\frac{1}{2}\right)$ would apply.

By using TFNs, the FAHP captures a range of possible importance levels rather than a single rating. In this way, even when the information is incomplete or experts have different points of view, the method remains flexible and accommodating. Once we start using these (l, m, u) triples (instead of single numbers), ordinary arithmetic no longer applies directly. Instead, we rely on fuzzy arithmetic, a set of fundamental operations designed for TFNs, which is specifically shown below.


**Multiplication**
For two TFNs $A=(l_{A},m_{A},u_{A})$ and $B=(l_{B},m_{B},u_{B})$,(2)$$A\times B\;=\;(\;l_{A}\times l_{B},\;m_{A}\times m_{B},\;u_{A}\times u_{B}\;).$$
**Exponentiation**
Raising a TFN *A* to a real power *p* yields(3)$$A^{p}\;=\;\left(l_{A}^{p},\;m_{A}^{p},\;u_{A}^{p}\right).$$
**Reciprocal**
For $A=(l_{A},m_{A},u_{A})$, the reciprocal of each bound is(4)$$A^{-1}\;=\;\left(\frac{1}{u_{A}},\;\frac{1}{m_{A}},\;\frac{1}{l_{A}}\right).$$
**Defuzzification**
Converting a TFN into a single value is often carried out by averaging its three bounds:(5)$$\mathrm{defuzz}\left(A\right)\;=\;\frac{l_{A}+m_{A}+u_{A}}{3}.$$

#### 2.3.2. Fuzzy Pairwise Comparison Matrix

In our FAHP framework, expert judgments on the relative importance of each criterion are captured in a fuzzy pairwise comparison matrix. Table 4 illustrates the fuzzy pairwise comparison matrix used for the criteria. Each cell (i, j) represents how much more important criterion i is compared to criterion j, expressed as TFNs in the [1, 9] range. Since the initial TFNs can be defined on an extended scale (e.g., values ranging from 1 to 15), a mapping is performed to normalize these fuzzy numbers to the standard AHP scale [1, 9]. The cumulative PD risk index is assigned the highest priority, underscoring its central role in indicating cable deterioration. The comparison further reveals that the number of PDs and age follow in importance, highlighting that cables with high PD occurrences and older aging are more vulnerable.

#### 2.3.3. Computation of Fuzzy Weights and Normalization

For each active criterion, its overall importance is determined by combining its fuzzy comparisons with all other criteria using the geometric mean. In other words, if a criterion *c* is compared against *n* other criteria, the geometric mean is computed as(6)$$\mathrm{GM}\left(c\right)={\left(\prod_{j=1}^{n}a_{c,j}\right)}^{\frac{1}{n}},$$
where $a_{c,j}$ represents the fuzzy number derived from comparing criterion *c* with criterion *j*.

Once the fuzzy weights are computed and defuzzified into crisp values w(c) for each criterion *c*, they are normalized to ensure that the sum of weights equals one. This is carried out by dividing each crisp weight by the total sum of all weights:(7)$$W\left(c\right)=\frac{w(c)}{\sum_{i=1}^{n}w\left(i\right)},$$
where W(c) denotes the normalized weight for criterion *c*, and the summation is taken over all *n* active criteria.

#### 2.3.4. Integration into Cable Vulnerability Score

After determining and normalizing the fuzzy weights through the FAHP, these weights are combined with the normalized performance measurements for each cable to derive a single vulnerability score. For each cable, each criterion (age, the number of PDs, etc.) is first normalized to ensure consistency across different scales. Then, each normalized value $x_{i}$ is multiplied by its corresponding normalized weight $W(c_{i})$. The vulnerability score is computed as the weighted sum across all active criteria:(8)$$\mathrm{Vulnerability}\;\mathrm{Score}=\sum_{i=1}^{n}W\left(c_{i}\right)\times x_{i},$$
where *n* represents the number of active criteria.

In other words, the final vulnerability score aggregates the contributions of all criteria (each adjusted by its relative importance derived from expert judgments) into a single metric. This systematic approach enables decision-makers to objectively rank cables and prioritize those at greatest risk based on a rigorously derived, transparent indicator. Finally, Figure 7 illustrates the overall structure of the fuzzy AHP cable vulnerability framework. Each step in the process, from fuzzification to ranking, is designed to handle uncertainty and incorporate subjective expert input in a mathematically consistent way. The use of fuzzy triangular numbers and normalized weight aggregation ensures both flexibility and interpretability, making the framework suitable for real-world utility applications.

Table 5 compares the pure fuzzy (rule-based) approach with the FAHP for cable vulnerability assessment. Both methods were implemented in this study; while the fuzzy method relies on linguistic rules to capture uncertainty, the FAHP incorporates pairwise comparisons and normalization to produce explicit weights and a crisp vulnerability score. The results and relative strengths of each approach are discussed later in the paper.

Figure 8 shows the framework used in this paper. It captures the interaction of all core modules, including data acquisition, uncertainty modeling, decision logic, and prioritization. It provides a comprehensive view of how the methodology transitions from raw data to actionable maintenance decisions. The process begins with the collection of data, such as cable age, PD characteristics, and the number of customers. Although distinguishing between customer types could offer added granularity in assessing failure impact and potential aging behavior, the current study adopts a uniform criticality assumption across all loads. Next, the fuzzy logic and FAHP setups are used. Monte Carlo simulations are utilized to take into account the ±10-year cable age uncertainty. Finally, the results are examined, and the cables are ranked based on their condition. This ranking scheme allows utilities to prioritize the maintenance of the most vulnerable cables, thereby improving grid reliability, reducing outages, and boosting customer satisfaction through a more reliable service.

## 3. Results

In this section, we present the outcomes of our cable vulnerability assessment. Since our PD sensor is not yet deployed, simulated PD data were used to test and refine our methods. Our current focus is on validating the methodology, with real-world data to follow once the sensor becomes operational. Both fuzzy logic and the FAHP were employed to calculate vulnerability scores for each cable and rank them for replacement. To rank the cables, three analyses were conducted: first, we used the age provided by the DSO as the actual age; second, as there was ±10-year uncertainty in these age values, we introduced this uncertainty to simulate potential inaccuracies; and third, we performed a correlation analysis on both the fuzzy logic and FAHP methods to evaluate their performance and robustness under these conditions. Our preliminary analysis suggests that these methods offer a more realistic assessment of cable condition by considering actual operational stresses and uncertainties rather than relying solely on age-based metrics.

### 3.1. Ranking Based on Reported Cable Age

In Table 6, the ranking of the first 45 cables (most of which have recorded PD events) based on three evaluation methods is shown. In addition, the data for each cable (including age, the number of PDs, the cumulative PD risk, and the number of customers) are provided. In the first view, differences arise among the three cable ranking methods. When cables are ranked solely based on age, those with similar ages have the same rank. This means that many cables share the same rank. This approach does not capture the nuances of cable condition, as it does not account for other influential factors that are not well characterized or commonly measured in traditional age-based strategies.

Let us examine the ranking of some specific cables to highlight the differences between the two evaluation methods. For example, cable “10476_10689”, which is 52 years old, with 1570 PD events and a cumulative PD risk of 1570, is ranked first by the FAHP but fourth by the fuzzy method. In contrast, cable “10370_10476” has a higher number of PD events (1940) but a lower cumulative PD risk (approximately 1358); it is ranked second by the FAHP yet first by the fuzzy method. Although both cables have high PD counts, cable “10476_10689” has greater PD severity, suggesting a higher likelihood of failure. In a practical sense, a cable with fewer but more severe PD events tends to be more susceptible than a cable with many low-amplitude PDs. Thus, in this scenario, the FAHP ranking better reflects the actual cable condition by balancing PD severity properly.

The same observation holds for cables “10103_10750” and “30086_30095”. Although these cables are younger than “10476_10738” and “10159_ 10689”, the severity of their PD events is higher, indicating a greater risk of failure. Consequently, the FAHP appropriately ranks these cables higher by accurately capturing the importance of PD severity, whereas the fuzzy method struggles to properly weigh one criterion against another. Numerous similar examples can be found in Table 6. Additionally, for cables without PD events, such as “30085_30100” (29 years old), the FAHP method provides a more logical ranking (34th) than the fuzzy method (72nd), and fuzzy logic incorrectly identifies younger cables without PD records as highly vulnerable. As previously noted, in the absence of PD data, cable age becomes a more critical factor, and the FAHP effectively incorporates this logic into its rankings.

In Figure 9, pairwise comparisons between the three evaluation methods for all 75 cables are illustrated. Each color represents a specific cable across all subplots, allowing for the visual tracking of rank changes between methods. The comparison between the FAHP and the age-based approach indicates that, while the FAHP produces rankings that differ due to its consideration of operational conditions, it still reflects age as an important factor. In contrast, the comparison between fuzzy logic and the age-based method suggests that fuzzy logic does not inherently prioritize one criterion over another. Instead, it relies on predefined rules for decision-making. The fuzzy method may struggle to distinguish between different cable conditions because these rules sometimes overlap. Although, in some instances, fuzzy rankings may outperform the classic age-based approach, doing so consistently requires a comprehensive and carefully defined set of rules. These differences arise from how each method incorporates the input indicators. The FAHP assigns explicit weights to criteria such as PD severity, count, and cable age, allowing it to highlight cables with fewer but more critical PD events or a higher aging risk. Although age remains an important index in all methods, the FAHP integrates it more effectively due to its capacity to balance it against other operational factors. In contrast, fuzzy logic lacks a built-in weighting mechanism, making it harder to consistently reflect the relative importance of age compared to other conditions.

### 3.2. Incorporating Age Uncertainty

Since the provided cable ages have an uncertainty of ±10 years, we apply a Monte Carlo simulation to account for this variation in both the FAHP and fuzzy logic models. Each cable is evaluated and ranked over 1000 simulation runs. This allows us to observe how the rankings shift under uncertain age values. As a result, we gain a better understanding of the distribution, stability, and sensitivity of the outcomes. This provides a clearer view of how each method performs when the input data are not fully precise.

Table 7 summarizes the average rankings from 1000 simulation runs, highlighting how the FAHP and fuzzy logic respond to age uncertainty and which cables consistently rank as most vulnerable. For cables that have PD records, the FAHP rankings remain unchanged in approximately 82% of the cases when compared to the baseline results in Table 6, where no age uncertainty is considered. This consistency highlights the greater influence of PD-related factors over age in determining cable vulnerability. It suggests that, even when exact age values are not available, the presence of reliable PD data allows the FAHP to produce stable and meaningful rankings.

In contrast, the consistency rate for the fuzzy logic approach is approximately 30%. While the remaining rankings are not identical, they tend to remain close to those obtained without considering age uncertainty. This means that, while fuzzy logic is more sensitive to input variability, it still has a relatively consistent pattern of ranking, particularly for cables with high condition indicators.

Figure 10 and Figure 11 show the mean rankings and corresponding uncertainty regions for all 75 cables through the FAHP and fuzzy logic methods with ±10-year age uncertainty. As shown in Figure 10, the FAHP approach maintains a narrow uncertainty band for most cables with PD records, demonstrating its robustness in scenarios where PD information is available. This aligns with the intended goal of sensor deployment, which is to capture meaningful PD data that strengthen decision-making. For cables with no PD records, the FAHP represents a wider level of uncertainty from its reliance on age and customer data. Since age is treated as a dominant criterion in the absence of PD information, the uncertainty in age propagates directly into the ranking outcome. In Figure 11, the fuzzy logic approach shows that the rankings for cables without PD records remain nearly unchanged. In contrast, cables with PD records exhibit wider uncertainty regions. As discussed in Section 3.1, fuzzy logic does not rank all the cables well, and it does not sufficiently prioritize the relative importance of the criteria. In general, the performance of fuzzy logic can depend on how clearly the input indicators are separated and how well the defined rules reflect expert understanding. When indicators overlap, or when multiple factors contribute similarly, capturing their influence accurately may require a more refined or extensive set of rules.

### 3.3. Robustness Analysis Through Correlation Metrics

In the third phase of our study, we explored how sensitive our cable vulnerability rankings were to uncertainties in the input data. For cables with PD records, we varied each parameter one at a time, i.e., adjusting the cumulative PD risk, the number of PDs, and the number of customers by ±20% while keeping the other values at their baseline. We then ran 1000 Monte Carlo iterations for each perturbed parameter and calculated the average ranking for each cable. This analysis allowed us to quantify the stability and reliability of the proposed framework under practical data variability. Finally, we computed the Spearman rank correlation between the baseline ranking (which is based on the age uncertainty ranking) and the rankings derived from each individual perturbation. The Spearman rank correlation was used because it compares the order of data points, making it effective for evaluating changes in cable rankings under uncertainty [38].

Figure 12 illustrates the FAHP-based cable rankings for cables with PD records, as derived from a comprehensive sensitivity analysis. In this analysis, each input criterion is individually perturbed by ±20% (while the remaining values are held constant). The results show that the rankings for cables with PD occurrences remain consistent across all perturbations. Furthermore, the Spearman correlation coefficients for the ranking comparisons are approximately 1.00, underscoring the robust performance of the FAHP method under input uncertainty.

The same sensitivity analysis is performed using fuzzy logic, with the results shown in Figure 13. In this case, cables with PD records are not clearly differentiated from those without, and many cables with PD events are assigned lower vulnerability scores than expected. Moreover, when each criterion is perturbed by ±20%, greater variability is observed in the rankings, and an average Spearman correlation coefficient of approximately 0.87 is obtained. This indicates that the fuzzy logic approach is less consistent in handling input uncertainty compared to the FAHP.

## 4. Discussion

A common limitation in asset management research is the inability to directly verify whether the highest-ranked cables have the shortest remaining life, as real-time failure observation would require long-term monitoring and risk, compromising network reliability. In this study, an indirect validation was provided through sensitivity analysis, comparative assessment with alternative methods, and alignment with physical PD indicators, all of which support the consistency and practical relevance of the proposed ranking framework.

In our cable vulnerability assessment, integrating condition-based inputs clearly enhances prioritization beyond traditional age-based methods. The proposed framework demonstrates that detecting only the peak amplitude of PDs provides critical insight into the actual condition of cables. The results indicate that the FAHP-based ranking framework reliably prioritizes input criteria. When there is no uncertainty in the input data, the FAHP method delivers logical and consistent rankings. Also, cable age is the dominant ranking when PD records are unavailable. In contrast, while the fuzzy logic approach has the potential to outperform classic age-based methods by incorporating expert knowledge, it requires an extensive and carefully calibrated rule set, particularly when the input criteria exhibit overlap.

In addition, the findings verify the validity of the FAHP approach to handling uncertainty in inputs. Through Monte Carlo simulation, ranking sensitivity was tested by perturbing cable age by ±10 years and the remaining input criteria by ±20%. As strong as this, the FAHP method in all cases continued to maintain constant ranks for cables with PD history, and their Spearman rank correlation coefficient values remained almost equal to 1.00. In contrast, the fuzzy logic approach was more inconsistent under similar conditions, with a mean Spearman correlation of approximately 0.87. This is an indication of reduced consistency under uncertainty conditions. These results indicate the practical reliability of the FAHP process for predictive asset management and confirm its comparative superiority to fuzzy logic in terms of generating consistency and rankings. The proposed framework is developed to be broadly applicable; however, practical implementation in other distribution networks should consider local cable characteristics and operational conditions, since the results obtained from the Danish network might differ from those in other settings. Previous cable vulnerability assessment methods in the literature include classical Multi-Criteria Decision-Making (MCDM) approaches such as the AHP and TOPSIS, along with various machine learning-based frameworks. Classical methods typically assume precise inputs and may inadequately capture the real-world uncertainties and expert subjectivity commonly encountered in distribution network data. Machine learning approaches, while highly effective in prediction and classification scenarios, generally face difficulties in explicitly addressing ranking-based problems, especially when labeled failure data are sparse, and the explicit prioritization of assets is required. In contrast, the FAHP and fuzzy logic methods introduced here seamlessly incorporate expert judgment and input uncertainty using fuzzy pairwise comparisons and membership functions. Furthermore, the robustness demonstrated by the FAHP method under input variations underscores its practical superiority in real-world utility applications, where input data often vary significantly in quality and completeness.

## 5. Conclusions

This study introduced a novel condition-based asset management framework for evaluating and ranking the vulnerability of underground cables using PD analysis, fuzzy logic, and the FAHP. Given the lack of a standardized ranking approach for such data, these decision-making methods were introduced. The ultimate goal is to support replacing the most vulnerable cables based on their actual condition rather than relying solely on chronological age. Traditionally, cable replacement has been age-based; however, many young cables may exhibit insulation degradation because of manufacturing defects, poor installation, or operating stress, which causes early PD activity. This is why PD is a sensible and early indication of cable aging, which allows utilities to react before complete failure. PD testing can be conducted online or offline, providing continuous condition assessment without de-energizing the circuit. The two PD surrogates employed (the event count and peak amplitude) capture complementary aspects of insulation degradation: the pulse count reflects defect density and severity, while the amplitude indicates the discharge energy associated with dielectric breakdown. Together, these features offer a validated and practically sufficient basis for ranking cable vulnerability at the network scale.

For PD records, the proposed framework only requires the peak values and number of PDs and not the complete time-series data to be saved. This facilitates an easier, low-cost, and memory-saving sensor with reduced computational demands. To support and validate the framework before sensor deployment, a real Danish distribution network is modeled using actual operational and environmental conditions, including soil temperature, moisture, and cable loading, to simulate synthetic PD events. The main objective is to evaluate and rank cable vulnerability based on PD severity, number of PDs, age, and affected customers.

Fuzzy logic and the FAHP are utilized in this research to prioritize cable vulnerabilities and facilitate condition-based and informed decision-making. While the FAHP allows for the systematic quantitative weighting of key criteria, fuzzy logic uses intuitive if–then rules to replicate expert thinking. To the best of our knowledge, the FAHP is utilized for the first time to prioritize the cable in asset management. The results demonstrate that, even with uncertainty (±10 years in age and ±20% in other factors), the FAHP constantly generates logical ranks with Spearman correlations near 1.00. Conversely, fuzzy logic exhibits more unpredictability, underscoring the advantages of the FAHP in terms of accuracy, stability, and expert-driven prioritization. The proposed methodology enables more targeted and data-driven cable replacement planning, which can help utilities avoid premature replacements and focus resources on the most at-risk assets. Although exact cost savings depend on the network size and investment strategies, condition-based prioritization can lead to a more efficient use of maintenance budgets and reduce unnecessary servicing and replacement efforts.

## Figures and Tables

**Figure 1 sensors-25-03454-f001:**
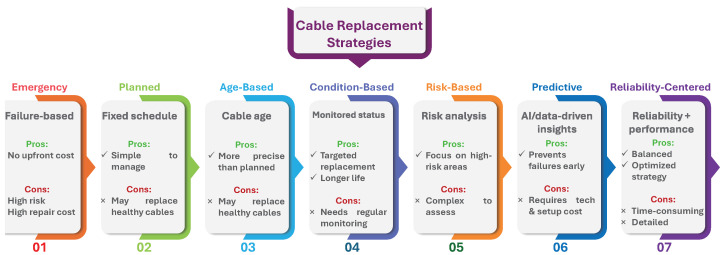
Cable replacement strategy categories and their key characteristics.

**Figure 2 sensors-25-03454-f002:**
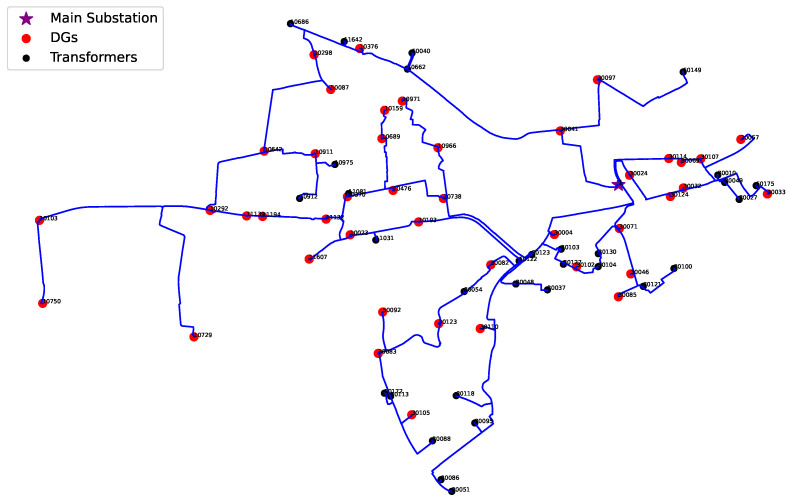
Schematic diagram of the utilized 10 kV Danish DSO grid.

**Figure 3 sensors-25-03454-f003:**
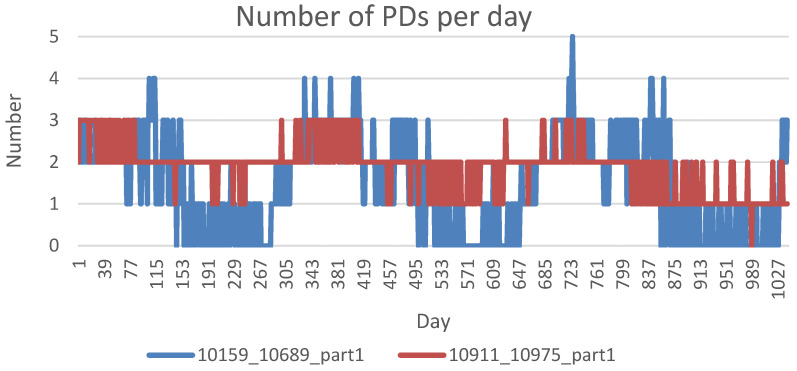
Estimated daily number of PDs over three years for two randomly selected cables.

**Figure 4 sensors-25-03454-f004:**
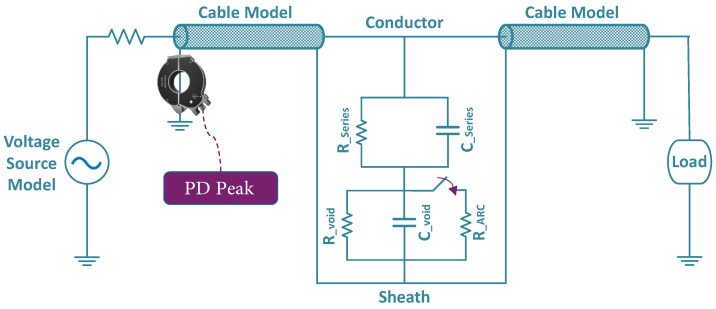
PD simulation model in DIgSILENT PowerFactory placed on underground cable shields based on [35]. The purple arrow represents the closing switch, which initiates the PD in the simulation.

**Figure 5 sensors-25-03454-f005:**
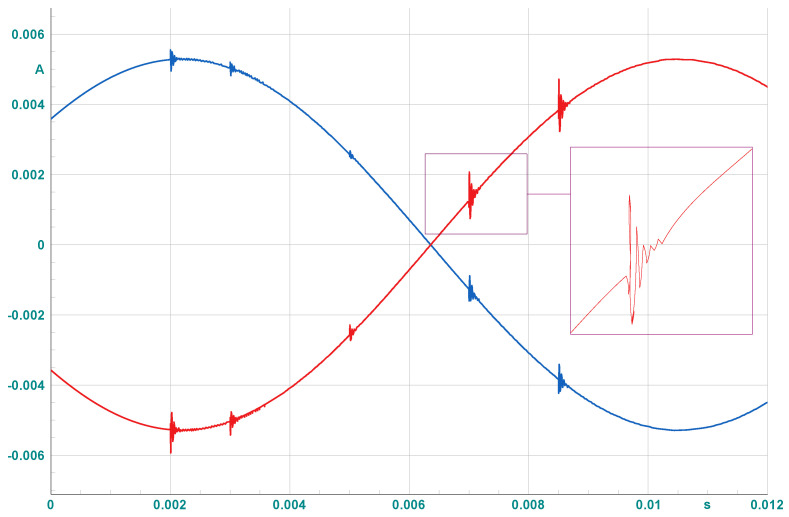
Example of replicated PD signals simulated within the software environment. Blue and red lines indicate PD signals measured at the two cable ends.

**Figure 6 sensors-25-03454-f006:**
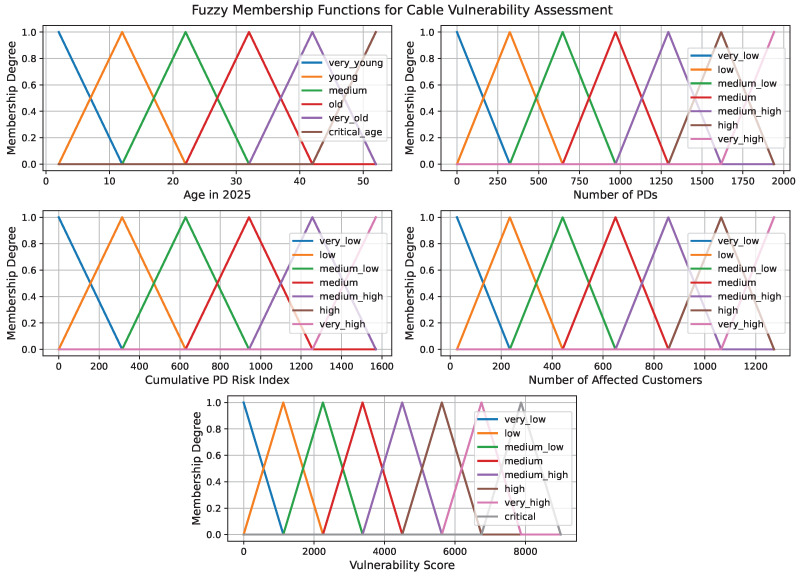
Triangular membership functions for input variables and output vulnerability score used in the fuzzy inference process.

**Figure 7 sensors-25-03454-f007:**
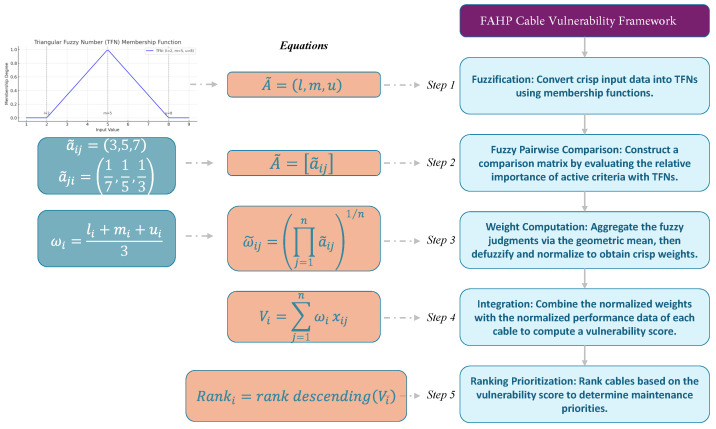
Overview of the FAHP cable vulnerability framework.

**Figure 8 sensors-25-03454-f008:**
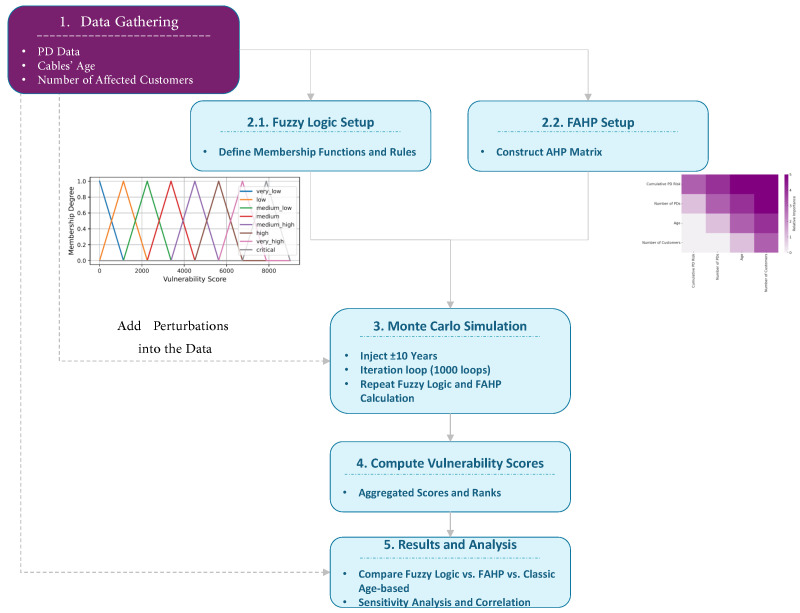
Cable vulnerability framework: data → fuzzy logic and FAHP → Monte Carlo simulation → cable ranking.

**Figure 9 sensors-25-03454-f009:**
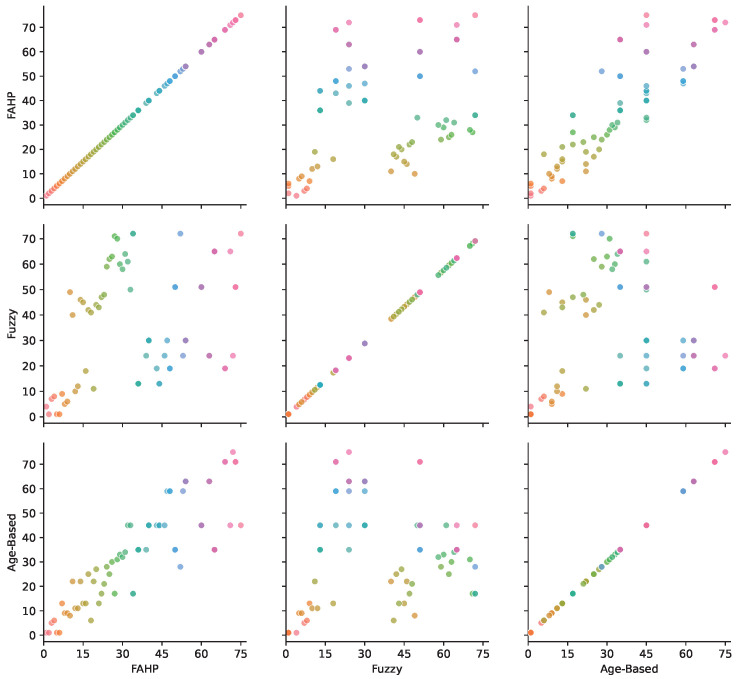
Pairwise ranking comparison among FAHP, fuzzy, and age-based methods.

**Figure 10 sensors-25-03454-f010:**
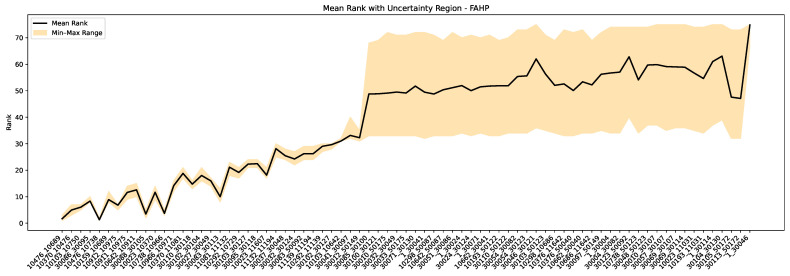
Cable rankings and uncertainty intervals from FAHP using Monte Carlo simulation (±10-year age uncertainty).

**Figure 11 sensors-25-03454-f011:**
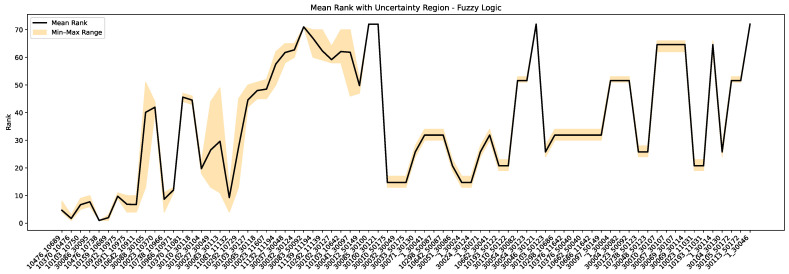
Cable rankings and uncertainty intervals from fuzzy logic using Monte Carlo simulation (±10-year age uncertainty).

**Figure 12 sensors-25-03454-f012:**
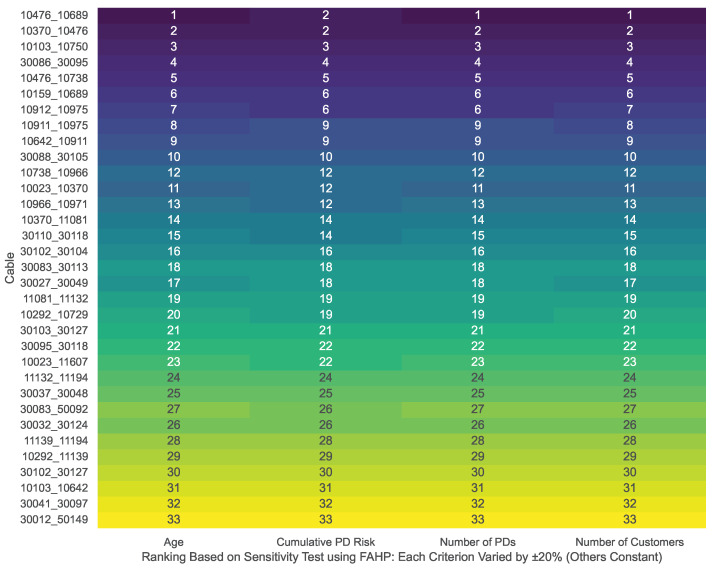
Robustness analysis of cable rankings: correlation outcomes from ±20% input variability for cables with PD records using FAHP.

**Figure 13 sensors-25-03454-f013:**
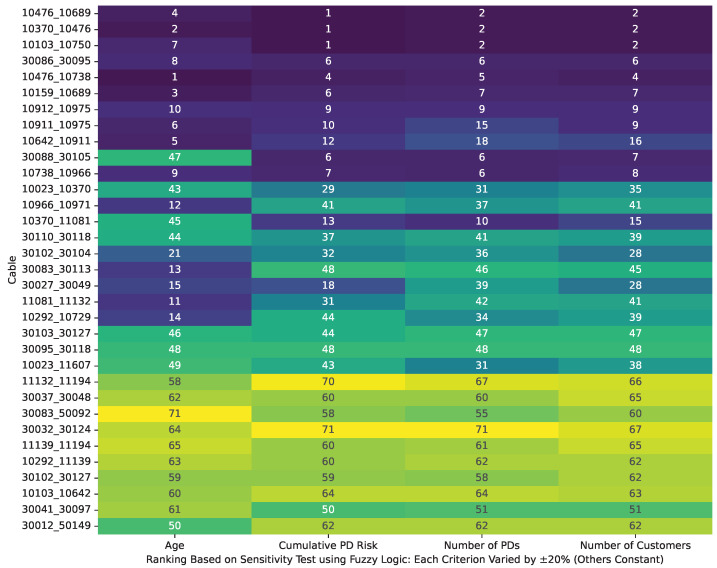
Robustness analysis of cable rankings: correlation outcomes from ±20% input variability for cables with PD records using fuzzy logic.

**Table 2 sensors-25-03454-t002:** Fuzzy rule conditions for estimating the number of PDs per day on each cable.

Rule Conditions	PD Probability
High Moisture & Old Cable Age & Low Soil Temperature & High Line Loading	High
High Moisture & Mid-Age Cable & Low Soil Temperature & High Line Loading	High
Medium Moisture & Old Cable Age & Low Soil Temperature & High Line Loading	High
High Moisture & Old Cable Age & Medium Soil Temperature & Medium Line Loading	High
Medium Moisture & Old Cable Age & Medium Soil Temperature & Medium Line Loading	Medium
Medium Moisture & Mid-Age Cable & Medium Soil Temperature & Medium Line Loading	Medium
Low Moisture & Old Cable Age & High Soil Temperature & Medium Line Loading	Medium
Medium Moisture & Mid-Age Cable & High Soil Temperature & Medium Line Loading	Medium
Low Moisture & Old Cable Age & High Soil Temperature & Low Line Loading	Low
Low Moisture & Mid-Age Cable & Medium Soil Temperature & Low Line Loading	Low
Medium Moisture & Mid-Age Cable & Low Soil Temperature & Low Line Loading	Low
Low Moisture & Low Soil Temperature & High Line Loading	Medium
Medium Moisture & High Soil Temperature & High Line Loading	High
High Moisture & Low Soil Temperature & Low Line Loading	Medium

**Table 3 sensors-25-03454-t003:** Severity levels of PDs based on cable type and discharge magnitude.

Severity Level	XLPE	XLPE Accessories	PILC ^1^	PILC Accessories
Discharge within acceptable range	0–250 pC	0–500 pC	0–2500 pC	0–4000 pC
Elevated risk, monitor frequently	250–500 pC	500–2500 pC	2500–7000 pC	4000–10,000 pC
Critical risk, immediate attention required	>500 pC	>2500 pC	>7000 pC	>10,000 pC

^1^ PILC: Paper-Insulated Lead Sheath Cable.

**Table 4 sensors-25-03454-t004:** Fuzzy pairwise comparison matrix for active criteria.

Criteria	Cumulative PD Risk	Number of PDs	Age	Number of Customers
**Cumulative PD Risk**	(1, 1, 1)	(2.14, 3.29, 4.43)	(3.29, 4.43, 5.57)	(5.57, 7.86, 9.00)
**Number of PDs**	(0.23, 0.30, 0.47)	(1, 1, 1)	(2.14, 3.29, 4.43)	(5.57, 6.71, 7.86)
**Age**	(0.18, 0.23, 0.30)	(0.23, 0.30, 0.47)	(1, 1, 1)	(4.43, 5.57, 6.71)
**Number of Customers**	(0.13, 0.18, 0.30)	(0.15, 0.18, 0.23)	(0.23, 0.30, 0.47)	(1, 1, 1)

**Table 5 sensors-25-03454-t005:** Comparison between fuzzy (rule-based) approach and FAHP.

Aspect	Fuzzy (Rule-Based)	FAHP
**Main Idea**	Uses *if–then* rules with fuzzy sets.	Relies on pairwise comparisons to assign fuzzy weights.
**Rules or Comparisons**	$m^{n}$*possible fuzzy rules*^1^.	$\frac{n(n-1)}{2}$ pairwise comparisons.
**Capturing Criterion Importance**	Must adjust membership sets or write rules giving certain criteria more weight.	Directly reflect importance in pairwise comparisons (e.g., “A is strongly more important than B”).
**Better For**	Problems needing flexible rule-based reasoning or smaller sets of criteria.	Problems requiring clear numeric ranking and structured weight derivation.

^1^*m* is the number of membership functions per criterion (low, medium, high)—*n* is the number of criteria (age, PD risk, …).

**Table 6 sensors-25-03454-t006:** Ranking comparison among FAHP, fuzzy, and age-based methods.

Line	Age	Number of PDs	Cumulative PD Risk	Number of Customers	FAHP	Fuzzy	Age-Based
10476_10689	52	1570	1570	1196	1	4	1
10370_10476	52	1940	1358	1196	2	1	1
10103_10750	44	1469	1469	1272	3	7	5
30086_30095	43	1461	1461	1256	4	8	6
10476_10738	52	1941	1073.7	1196	5	1	1
10159_10689	52	1916	1002.2	1196	6	1	1
10912_10975	35	1341	1341	1272	7	9	13
10911_10975	39	1446	1012.2	1272	8	5	9
10642_10911	39	1406	984.2	1272	9	6	9
30088_30105	42	1074	1074	1061	10	49	8
10023_10370	27	1083	1083	1256	11	40	22
10738_10966	36	1300	910	1196	12	10	11
10966_10971	36	1253	877.1	1196	13	12	11
10370_11081	27	978	978	1256	14	46	22
30110_30118	35	897	897	1256	15	45	13
30102_30104	35	817	817	1196	16	18	13
30027_30049	25	753	753	1238	17	42	25
30083_30113	43	835	584.5	1061	18	41	6
11081_11132	27	903	632.1	1256	19	11	22
10292_10729	22	736	736	1256	20	44	27
30103_30127	35	742	519.4	1196	21	43	13
30095_30118	29	711	497.7	1256	22	47	17
10023_11607	28	709	496.3	1256	23	48	21
11132_11194	21	507	507	1256	24	59	28
30037_30048	25	436	436	1196	25	62	25
30032_30124	19	440	440	1238	26	63	30
30083_50092	29	464	364	1061	27	71	17
11139_11194	18	463	324.1	1256	28	70	31
10292_11139	15	351	351	1256	29	60	33
30102_30127	17	305	305	1196	30	58	32
10103_10642	11	254	177.8	1272	31	64	34
30041_30097	7	248	57.1	1272	32	61	45
30012_50149	7	112	78.4	1272	33	50	45
30085_30100	29	0	0	26	34	72	17
30100_30121	29	0	0	26	34	72	17
30010_50175	8	0	0	1238	36	13	35
30032_30049	8	0	0	1238	36	13	35
30033_50175	8	0	0	1238	36	13	35
30071_30130	8	0	0	1196	39	24	35
1_30041	7	0	0	1272	40	30	45
10298_50087	7	0	0	1272	40	30	45
10642_50087	7	0	0	1272	40	30	45
30051_30086	7	0	0	1256	43	19	45
1_30024	7	0	0	1238	44	13	45
30024_30124	7	0	0	1238	44	13	45

**Table 7 sensors-25-03454-t007:** Average cable rankings using FAHP and fuzzy logic based on Monte Carlo simulation.

Line	Age	Number of PDs	Cumulative PD Risk	Number of Customers	FAHP	Fuzzy
10476_10689	52	1570	1570	1196	1	4
10370_10476	52	1940	1358	1196	2	2
10103_10750	44	1469	1469	1272	3	7
30086_30095	43	1461	1461	1256	4	8
10476_10738	52	1941	1073.7	1196	5	1
10159_10689	52	1916	1002.2	1196	6	3
10912_10975	35	1341	1341	1272	7	10
10911_10975	39	1446	1012.2	1272	8	6
10642_10911	39	1406	984.2	1272	9	5
30088_30105	42	1074	1074	1061	10	47
10738_10966	36	1300	910	1196	11	9
10023_10370	27	1083	1083	1256	12	43
10966_10971	36	1253	877.1	1196	13	12
10370_11081	27	978	978	1256	14	45
30110_30118	35	897	897	1256	15	44
30102_30104	35	817	817	1196	16	21
30083_30113	43	835	584.5	1061	17	13
30027_30049	25	753	753	1238	18	15
11081_11132	27	903	632.1	1256	19	11
10292_10729	22	736	736	1256	20	14
30103_30127	35	742	519.4	1196	21	46
30095_30118	29	711	497.7	1256	22	48
10023_11607	28	709	496.3	1256	23	49
11132_11194	21	507	507	1256	24	58
30037_30048	25	436	436	1196	25	62
30083_50092	29	464	364	1061	26	71
30032_30124	19	440	440	1238	27	64
11139_11194	18	463	324.1	1256	28	65
10292_11139	15	351	351	1256	29	63
30102_30127	17	305	305	1196	30	59
10103_10642	11	254	177.8	1272	31	60
30041_30097	7	248	57.1	1272	32	61
30012_50149	7	112	78.4	1272	33	50
30100_30121	29	0	0	26	34	72
30085_30100	29	0	0	26	35	72
30010_50175	8	0	0	1238	36	16
30033_50175	8	0	0	1238	37	16
30032_30049	8	0	0	1238	38	16
10298_50087	7	0	0	1272	39	33
1_30041	7	0	0	1272	40	33
10642_50087	7	0	0	1272	41	33
30051_30086	7	0	0	1256	42	22
1_30024	7	0	0	1238	43	16
30024_30124	7	0	0	1238	44	16
30071_30130	8	0	0	1196	45	27

## Data Availability

The data presented in this study are not publicly available due to privacy and confidentiality restrictions. Further inquiries can be directed to the corresponding author.

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
