# Peer review of "Partial Discharge-Based Cable Vulnerability Ranking with Fuzzy and FAHP Models: Application in a Danish Distribution Network"

_sensors, 2025, doi:10.3390/s25113454_

Round 1
Reviewer 1 Report
Comments and Suggestions for Authors
In this paper, a partial discharge-based cable vulnerability ranking method has been designed by using fuzzy theory and FAHP model, and an example about Danish distribution network has been analyzed. Some revise should be done:
- The title should add a word ‘logic’ and can be revised it as ‘A partial discharge-based cable vulnerability ranking method based on fuzzy logic and FAHP model and application in Danish distribution network’
- The abstract should rewrite to show the four innovations of this paper more clearly.
- The abstract should be given the comparison results in the last sentence.
- The four key contributions should be described more specifically, and it can be written as four long paragraphs.
- The designed method should be explored more and more detailed. The Figure 7 and Figure 8 are both too simple.
- The difference reasons for Figure 9 should be added.
- The comparison research work should be added in Section 4.
Author Response
Dear Reviewer,
Please find our detailed responses to your comments in the attached file. We appreciate your valuable feedback and the opportunity to improve our manuscript.
Best regards,

Reviewer 2 Report
Comments and Suggestions for Authors
This is a good paper that deserves publication.
It would be strengthened if the authors made the following points more clearly.
- There is no well-established physics of failure model for power cables. So, the replacement approaches are based on presumed surrogates for remaining life.
- This paper uses a plausible set of surrogates. But there is no proof that the set used is complete or appropriate.
- While it is plausible, and even likely, that this approach provides improved cable replacement guidance, there is no way to validate that the cables replaced via this approach had the shortest remaining life.
Specific recommendations:
Line 4 – Faults are not initiated by partial discharges. They are related to the electrical, mechanical, thermal, and chemical environment in which the insulation operated. Incipient faults can sometimes be indicated by partial discharges. In addition, the temperature and pressure rise, and the plasma etching caused by a partial discharge can accelerate the rate of aging of the fault.
Line 5 – The trends promise to provide valuable information, but there is no reliable approach to extracting age information today.
Line 32 – While the sentence is often true, cables can age and fail with little or no partial discharge activity.
Line 40 – I doubt that you really meant to say, “long kilometers.”
Line 89 – It was important to highlight that there is unlikely a linear correlation between partial discharge behavior and remaining life or other parameters.
Line 173 – Providing the uncertainty in the time of service and using a monte Carol approach to explore this uncertainty add to the paper’s quality.
Line 193 – Should be “ends” not “end”.
Line 198 – If you assert no partial discharges, you must also identify the smallest partial discharge that would be detected.
Line 227 – The paper’s quality is enhanced by the acknowledgment that time in service is but one indicator of remaining life. Arrhenius’ law tells us that a higher operating temperature can accelerate aging significantly.
Line 327 – It is prudent to try to reduce the results of a complex analysis to a single number. The desire for the number appears to be a strong human desire because important decisions across multiple fields and globally are based on the value of a single real number.
Line 338 – the number of customers is a good surrogate for estimating the impact of failure. But it naturally triggers the realization that the types of customers may have a significant correlation with the remaining life.
Line 364 – The paper quality is improved by this observation. But “overlooks” may not be the right verb as it implies that the other factors were well know but ignored. It is more accurate to observe these factors are not well known or characterized.
Line 479 – Here it should be noted that the approach is expected to be general but would require correlation with local experience. There is no reason to expect that the Danish cable environment is typical.
Author Response

(The authors gave the same response as above.)

Reviewer 3 Report
Comments and Suggestions for Authors
The topic of this article is relevant and aimed at reducing the cost of replacing electrical cables and ensuring reliable operation. The list of references contains 37 links and complete and sufficient for an article of this type. This article is concise and well structured, the division into sections consistently and logically sets out the essence of the matter.
In my opinion, the main disadvantage of this paper [page 6, line 175] is the use of cable line simulation for partial discharges. This means a study without an experimental part. All results are obtained using simulation and statistical data.
To improve this paper:
[page 4, line 87] please add description of the “pC” units (looks like “pF” of capacitance), for the first appearance it will be unclear to readers
[page 6, line 173] please extend description of “±10-years uncertainty” compared to real age based on data from DSO (what is the newest cable? Older than 10 years?)
[page 21, line 499, Conclusion] please add description about - is it possible to reduce the costs of servicing cable lines using the proposed methodology? how much cheaper?
I recommend this article after minor revision.
Author Response

(The authors gave the same response as above.)

Round 2
Reviewer 1 Report
Comments and Suggestions for Authors
All questions have been revised and can be acceptted for publish.